# Heart Failure in Patients with Arrhythmogenic Cardiomyopathy

**DOI:** 10.3390/jcm10204782

**Published:** 2021-10-19

**Authors:** Shi Chen, Liang Chen, Firat Duru, Shengshou Hu

**Affiliations:** 1Department of Cardiac Surgery, State Key Laboratory of Cardiovascular Disease, Fuwai Hospital, National Center for Cardiovascular Diseases, Chinese Academy of Medical Sciences and Peking Union Medical College, 167A Beilishi Road, Xi Cheng District, Beijing 100037, China; chenshi@fuwai.com (S.C.); huss@fuwaihospital.org (S.H.); 2Arrhythmia and Electrophysiology, Department of Cardiology, University Heart Center Zurich, Rämistrasse 100, CH-8091 Zurich, Switzerland; firat.duru@usz.ch; 3Center for Integrative Human Physiology, University of Zurich, Rämistrasse 100, CH-8091 Zurich, Switzerland

**Keywords:** arrhythmogenic ventricular cardiomyopathy, heart failure, risk stratification, prognosis

## Abstract

Arrhythmogenic cardiomyopathy (ACM) is a rare inherited cardiomyopathy characterized as fibro-fatty replacement, and a common cause for sudden cardiac death in young athletes. Development of heart failure (HF) has been an under-recognized complication of ACM for a long time. The current clinical management guidelines for HF in ACM progression have nowadays been updated. Thus, a comprehensive review for this great achievement in our understanding of HF in ACM is necessary. In this review, we aim to describe the research progress on epidemiology, clinical characteristics, risk stratification and therapeutics of HF in ACM.

## 1. Introduction

Arrhythmogenic cardiomyopathy (ACM) is a familial heart disease with a prevalence of approximately 1:5000 [1,2,3]. The disease is a major cause of sudden cardiac death (SCD) in adolescents and young adults, especially during athletic activity. With the establishment of clinical risk prediction models and the use of implantable cardioverter-defibrillators (ICD) in high-risk patients, the incidence of malignant arrhythmic events has gradually reduced [4,5,6,7,8]. ACM is known to be a progressive disease, and with advanced right ventricular (RV) involvement and/or left ventricular (LV) involvement, symptoms and signs of heart failure (HF) may occur during later stages of disease. Despite the fact that the focus in ACM was on arrhythmias in earlier studies, various HF phenotypes have received attention more recently.

The prevalence and severity of progressive HF in ACM have been somewhat controversial, as the epidemiological and clinical characteristics of the disease varied greatly among reports from different centers [9,10,11,12]. In many retrospective clinical studies, HF was reported to be rare and often related to later stages of disease [13,14]. On the other hand, HF was also reported to be an early, and even first, manifestation of disease in other studies. In particular, it has even been recognized as one of the main causes for cardiac death and heart transplantation (HTx) [10,12].

The HF course of ACM is unique from other cardiac diseases, such as dilated cardiomyopathy and hypertrophic cardiomyopathy, which are more common causes for severe HF related adverse outcomes. The sequence and origin of HF symptoms in ACM is quite distinctive. The symptoms of HF often appear after the stage of electrophysiological disorder and originate from the right side pulmonary circulatory system in ACM. The severity of ACM has gender specificity and age differences, which could lead to different risk stratification from other cardiomyopathies. Apart from that, the genotype and exercise could also affect the HF progression in ACM, which requires a more specific prevention and treatment strategy in its clinical management.

In this review, in order to provide a more comprehensive understanding of clinical phenotypes and management strategies, we report the prevalence, clinical course, risk stratification, prevention and treatment strategies of HF in ACM patients. Moreover, we propose our view of prospective research directions in this field.

The ACM studies included in this review were found on PubMed, Embase. We combined arrhythmogenic right ventricular cardiomyopathy/dysplasia OR ARVC/D (arrhythmogenic right ventricular cardiomyopathy/dysplasia) with heart failure OR cardiac dysfunction OR cardiac insufficiency as keywords and MeSH terms, and manually searched the reference lists of key reviews and all potentially relevant studies. The search was performed from its inception until 15 September 2021.

## 2. Prevalence of HF in ACM

Progressive disease and occurrence of HF was described in various ACM registries across different ethnic backgrounds. HF has been reported from all ACM cohorts across the globe (Figure 1). However, the prevalence of HF in this disease was reported differently in different centers because of the lack of consensus on the definition of HF and due to different inclusion methods. The clinical diagnosis of HF was mainly based on a combination of the American College of Cardiology/American Heart Association HF staging system and physician experience in different registries. The incidence of HF was reported to be in the range of 5% (defined by volume overload) to 49% (defined by the clinical symptoms and the severity of ventricular remodeling in echocardiogram or cardiac MRI imagines) among different centers and studies [10,15]. In the clinical reports from studies which only include the ACM probands, the incidence of HF was higher compared to some others which also enrolled their at-risk relatives. The incidence of HF in ACM was reported to be around 0.5% annually in primary care hospitals, which was much lower than that reported by tertiary care centers [16]. The causes of this difference may mainly derive from the unavoidable selection bias of the ACM population, in that patients with advanced HF were more likely treated in tertiary hospitals.

The global incidence of adverse outcomes, including HTx and cardiac death caused by HF in ACM, was reported to be 2 to 22% (Table 1). Advanced HF in ACM was more common in Asia and could cause a poorer prognosis compared with Europe or America. The ACM patients from China or Japan had higher risk for HF related rehospitalization, heart transplantation and death. This could possibly be caused by the different proportion of certain gene mutations, such as *plakophilin 2* (*PKP2*) and *Desmoglein 2* (*DSG2*).

## 3. Clinical Characteristics and Classification

### 3.1. Clinical Characteristics

#### 3.1.1. Clinical Course and Symptoms

Electrical instability and progressive HF are the typical phenotypes of ACM. The symptoms such as fatigue, dyspnea, and edema are often caused by advanced HF [15]. Symptoms of HF may occur at any age, ranging from childhood to the elderly [26,29]. However, later-onset HF was more common in ACM. The prognosis was worse in these patients than other ACM patients who only had arrythmias. The average age of HF presentation was 40 to 46 years, which was approximately 10 years later than occurrence of arrhythmic events, in general [10,30]. The mortality of patients who had first manifestation with HF symptoms was higher than patients who only had ventricular tachyarrhythmias. The risk of adverse outcomes increased significantly if patients were rehospitalized for HF [38].

The clinical course of HF in ACM was reported to be heterogeneous. Some ACM patients reached the endpoint of cardiac death or HTx within 2 or 3 years, while these occurred after a few decades in some patients [15,29,30]. The description of HF’s course in ACM was limited in most studies and the potential relationship between HF and ventricular arrhythmias was incompletely described [39].

In some opinions, HF and ventricular arrythmias were two independent phenotypes, as the incidence of VT was common for both, and it made no difference whether patients had HF or not [18,40]. Besides, clinical management of VT, such as ICD implantation and catheter ablation, couldn’t reduce the occurrence of HF and improve its prognosis in ACM [10,12,27,31,32,34,35]. In other points of view, however, HF and ventricular arrythmias occurred successively based on the four sequential clinicopathological stages of the classical ACM (Figure 2) [13,41].

Stage 1, the concealed stage, characterized by minor structural changes in RV without obvious abnormality in electrocardiogram (ECG), echocardiogram and histological findings. At this stage, SCD and life-threatening ventricular arrythmias can be the first manifestation in young patients, especially if they are engaged in competitive sports and endurance exercise.Stage 2, electrical instability stage, in which RV structural remodeling and dysfunction become an overt phenotype. Recurrent RV arrhythmias such as VT frequently occurred at this stage.Stage 3, RV failure stage, caused by diffuse progressive fibrofatty tissue replacement of RV myocardium in RV. LV function is typically preserved. Symptoms of volume overload and congestive HF appear gradually but are still under control, if proper intervention strategies are used at this stage.Stage 4, biventricular HF with global dilation and LV involvement. The proportion of ACM patients to reach this phase was small, which could be influenced by survival bias.

In summary, arrhythmias typically occur in the early phase of ACM, whereas progressive RV failure and LV dysfunction may appear during the later natural course of disease. The clinical course of HF is typically quite different in ACM as compared to other common cardiomyopathies, such as dilated cardiomyopathy, and has unique gene expression patterns. This clinical classification and the stages of HF were mainly classified based on the natural course of classical ARVC, however, the natural history of other ACM subtypes such as LV dominant ACM still need further study.

#### 3.1.2. Imaging and ECG Phenotype


The ARVC patients could have a typical phenotype with both structural abnormalities and electrophysiological disorders, which are often reflected on the results of imaging and ECG tests. The typical CMR and ECG characteristics of ACM HF patients are provided in Figure 3.

The diagnosis of ARVC is determined by the combination of imaging, electrophysiological, pathological examination and family history, based on the 2010 Revised Task Force Criteria (TFC) [43]. The diagnostic criteria of imaging and ECG in 2010 TFC are summarized in Figure 4.

Cardiac imaging modalities, including echocardiogram and cardiac magnetic resonance (CMR), are reliable methods to make a diagnosis of ACM and reflect the progression of HF [44]. The echocardiogram examination cannot evaluate the fibrofatty replacement of the myocardium wall and has limited power to precisely measure right ventricular cardiac function [45]. Thus, further CMR examination is the basic modality to make a definite diagnosis of ACM. Apart from the critical role in ACM diagnosis, the abnormalities in echocardiogram and CMR can also indicate the prognosis of HF in ACM patients. The echocardiography presentations in ACM patients with reduced LVEF and tricuspid regurgitation are risk factors for HTx and cardiac death [18,22]. As for CMR parameters, in one report from a prospective study [46] which compared the characteristics of normal controls and ACM patients, significant right ventricular ejection fraction (RVEF) reduction, LV end-diastolic diameter/LV end-systolic dimension (LVEDD/LVEDS) increase, LV global and regional peak strain impairment and higher prevalence of late gadolinium enhancement (LGE) were observed in ACM patients with reduced LVEF. A greater amount of LGE in LV was mainly localized in the subepicardial wall layers, which also negatively correlated with LVEF [47]. The specific prognostics role of these abnormalities in CMR images in predicting adverse outcomes of HF is still not clarified. However, they could be tested as imaging risk factors for end-stage HF in the future.

ECG abnormalities, including depolarization and repolarization abnormalities and arrhythmias, can be specific characteristics of ACM [48], which are also among the major diagnostic criteria [43]. Distinct from HF in dilated cardiomyopathy, ACM patients with HF have a higher proportion of low QRS voltages in limb leads, T-wave inversions in the inferolateral leads and major ventricular arrhythmias. Low 12-lead QRS voltage is an independent indicator for heart transplantation in ACM [49]. The inferior leads TWI, a precordial QRS amplitude ratio of ≤ 0.48 and QRS fragmentation are correlated with adverse outcomes, including malignant arrythmias and heart failure events [50]. Additionally, first-degree atrioventricular block and epsilon waves are among the predictors for HF hospitalization [38].

#### 3.1.3. Plasma Biomarkers

Plasma biomarkers, which can reflect the severity of HF and, to some degree, predict prognosis, are often used in the clinical practice. According to the findings of ACM cohort studies, biomarkers such as various plasma proteins, noncoding RNA and autoantibodies may be potentially correlated with adverse cardiovascular events, including malignant arrythmias and end-stage heart failure outcomes (Figure 5). In this review, we mainly discuss the application of the plasma biomarker in HF prediction. BNP and NT-proBNP, which are recommended as gold-standard biomarkers for HF by guidelines, can also be applied in HF management. The increase in NT-proBNP is correlated with RV dilation and cardiac dysfunction in ACM [51,52]. Another biomarker as cardiac troponin I (cTnI), which could indicate for cardiomyocyte death and cardiac muscle injury and also correlate with premature ventricular contractions in ACM patients. These classical cardiac biomarkers can reflect the severity of HF in some degree. However, the specificity of NT-proBNP, cTnI and other widely used cardiac biomarkers is low, and their application in ACM patients is still controversial.

In a proteome study, heat shock protein 70 (HSP70) was significantly elevated in both ACM and other cardiomyopathies [53]. Furthermore, bridging integrator 1 (BIN1) [54,55], ST2 [56] and galectin-3 (GAL-3) [57] can reflect the severity of HF in ACM. The circulation level of complements is also correlated with mortality and cardiac dysfunction in ACM [58]. The sC5b6 level could display the severity of HF, which is significantly higher in ACM patients with biventricular dysfunction compared with isolated RV dysfunction. As is well-known, the abnormality of lipid metabolism is one of the key pathogenesis for ACM progression. The oxidized low-density lipoprotein (ox-LDL) could not only reflect the severity of ACM fat infiltration, but also predict the HF and malignant arrhythmia’s event risk. Another specific plasma biomarker for ACM clinical course discrimination and cardiac function prediction is β-hydroxybutyrate (β-OHB). In the report from Fuwai Hospital, the level of β-OHB is relatively low in healthy volunteers and unsuspected relatives of ACM patients. However, it is gradually increased in ACM patients following the clinical heart failure stage from normal cardiac function to isolated RV dysfunction and biventricular cardiac dysfunction. It could be recognized as a useful predictor for disease progression and adverse heart failure outcomes.

These findings were all from small-sized cohorts with limited numbers of ACM patients with HF, and were not validated by others. Such biomarkers are also not specific for ACM, as end-stage of HF is the common pathway in various cardiomyopathies. The serum levels of these biomarkers are not changed significantly at an early stage and are only evaluated since the onset of advanced HF, which limits their application in early diagnosis and guidance on early intervention. Thus, further proteomics and metabolomics studies on HF biomarkers in ACM are still needed.

#### 3.1.4. Histopathological Characteristics and Endomyocardial Biopsy

The hearts from orthotopic HTx and/or autopsies in ACM patients had distinctive histopathological features as compared to other cardiomyopathies. The typical gross morphology and histopathological characteristics of ARVC HF patients are provided in Figure 6.

The fibrofatty replacement and cardiomyocyte death in the ventricular myocardium are classical pathological changes in ACM. These histopathological changes commonly start from the RV epicardium, gradually progress into subendocardial layers and demonstrate transmural extension [59]. The endomyocardial biopsy (EMB) is a histopathological evaluation methods to make a definite diagnosis of ARVC, recommended by 2010 TFC and excluding sarcoidosis, myocarditis or other phenocopies that could also lead to uncontrollable HF [3]. However, considering that the sensitivity of EMB was low and the myocardium free wall of RV was thin, its application in ACM patients with HF is still controversial. From the recent study, the EMB is safe and helpful for further enhancing the diagnostic efficiency to the arrhythmogenic left ventricular cardiomyopathy, which posed a higher risk to HF stage progression [60]. More evidence will be needed to confirm the clinical value of EMB in the suspension ACM patients.

### 3.2. Clinical Classification

The clinical course and phenotype of ACM is dynamic and distinctive in an individual patient. ACM phenotype was typically divided into four distinct clinical stages [61]. With deeper understanding of the impact of genotype on clinical manifestations, a novel classification (Fuwai Classification) has been recently proposed by our group (Table 2) [42]. In this classification, ACM patients who carried desmosomal mutations, including *DSG2*, *desmocollin 2* (*DSC2*) and *PKP2* in cluster 1, tended to have early onset of HF and reached the endpoint of cardiac death or HTx in a shorter period compared with other subtypes.

## 4. Risk Factors and Stratification

The risk factors for HF in ACM include demographics factors, electrophysiological abnormalities, genotype and physical activity. The certain genes mutations of ACM patients such as *DSG2*, *desmoplakin* (*DSP*), and *phospholamban* (*PLN*) have been proven to be one of the fundamental causes for processive HF, while the factors such as physical activity and arrhythmias often induce or aggravate the HF’s progression. However, the role of other malignant arrythmia risk factors in ACM HF such as gender, age or family history might still be controversial (Figure 7).

### 4.1. Demographics

#### 4.1.1. Gender

It is well known that gender and sex hormones are important risk factors in ACM. The prevalence of ACM is significantly higher and malignant arrhythmias are often more severe in male patients. However, the HF risk in different genders is still controversial in ACM. [63] According to most reports, there was no obvious gender differences in HF’s incidence or severity [28,63,64,65]. The manifestations of the ECG and echocardiogram were also similar in patients with HF regardless of gender. The combined Johns Hopkins/Netherlands cohort study suggested that the risk of LV involvement and adverse outcomes caused by HF had no difference among the two genders. However, in one study, the risk for cardiac death and HTx caused by HF were reported to be higher in male patients, which was explained by the more competitive exercise patterns in these patients [11]. In contrast, another study found that female sex was an independent risk factor for HF [38]. A potential explanation for this observation was that poor endurance of volume overload and smaller thoracic size often occurred in female patients [64,65]. The different role genders play in HF progression among these studies may derive from the balance between the negative effect of sex hormones such as testosterone on cardiomyocytes apoptosis and lower tolerance to HF due to the physiological characteristics of female sex. The limited patient size can also lead to potential selective bias and an impact on the general results in the real world.

#### 4.1.2. Age

The average age of onset for ACM is 30–40. It is widely accepted that early-onset of HF often indicates more rapid occurrence of adverse outcomes. On the other hand, in a small cohort with young ACM patients under 18 years and with long-term follow-up, the incidence of adverse HF outcomes was similar in children and adults [26]. In the report from the Nordic ACM Registry, an age of disease onset under 35 years was shown to be an independent risk factor for HTx [30]. In contrast, older patients with late presentation of ACM, who were thought to be more vulnerable, had less risk of cardiac death and HTx than younger patients [29].

#### 4.1.3. Family History and Ethnic Background

There is widespread consensus that family history is an important risk factor for ventricular arrhythmic events in ACM patients. ACM probands who had symptomatic relatives have a higher possibility for carrying desmosomal mutation and a higher risk of SCD [66]. With respect to the risk of HF, however, most reports showed that familial or isolated ACM probands had the same risk of developing HF [14,19,30]. Meanwhile, no evidence showed that ethnic background could affect the HF risk in ACM patients [10].

### 4.2. Arrhythmias and Electrophysiological Abnormalities

Arrhythmias may occur at any stage of ACM, and over 90% of patients had an initial manifestation or the history of VT, atrial arrhythmia or aborted SCD [13,67,68]. Many studies demonstrated the critical role of arrhythmias in the progression of HF. Theoretically, patients with recurrent VT and atrial arrhythmias tended to have worse hemodynamics features and pump dysfunction [23,32,39]. However, there was no evidence suggesting that VT is a risk factor for HF progression. In addition, the right bundle branch block (RBBB) was reported to have a significant correlation with severe biventricular HF [17]. The development of complete RBBB may lead to poor prognosis [69].

Several studies focused on the characteristics of atrial arrhythmias in ACM and found that these arrhythmias may be common, with a prevalence of about 10–20% in the general ACM population [19,30,39]. The incidence of atrial fibrillation in patients with adverse HF outcomes was 6–19% [30]. Atrial arrhythmias were associated with increasing mortality and morbidity and appeared to be risk factors for HTx and deaths due to end-stage HF. In addition, the incidence of first-degree atrioventricular block may be significantly higher in patients with rehospitalization for HF, and it was shown to be an independent risk factor for HF hospitalization in ACM [38].

ECG abnormalities may also provide useful information to predict prognosis due to HF in ACM. Prolonged PR intervals, prolonged QRS in lead V1, T wave inversion in leads V4–V5–V6, epsilon waves, presence of bundle branch block and low QRS voltages were reported to be potential risk predictors for adverse outcomes in ACM [19,27]. However, the majority of these studies set their end-point as both lethal arrhythmic events and HTx/death due to HF. A study focusing only on HF outcomes suggested that the precordial QRS amplitudes may be an indicator of RV remodeling and have the potential to predict the progression of HF [68]. Another study of HF in ACM found that the presence of negative T waves in precordial leads V4–V6 was more common in patients with HF symptoms [10]. In addition, the incidence of epsilon waves was significantly higher in patients with HF hospitalization [38].

### 4.3. Genotype

It is well-recognized that ACM is an inherited cardiomyopathy mainly caused by desmosomal mutations. The genotype may also impact the likelihood of developing significant HF during the disease course. Patients carrying *DSG2* gene mutations more often demonstrate HF progression compared with *PKP2* carriers [9]. Previous studies suggested that patients carrying *DSG2* gene mutations tended to have lower LVEF [70]. ACM patients carrying homozygous p. Phe531Cys variant in *DSG2* commonly develop severe LV dysfunction and biventricular failure at a young age [71]. *DSP* and *PLN* mutations are also correlated with LV involvement [10,11,13,36,72,73,74,75,76,77,78,79,80,81]. The risk of irreversible HF in desmosomal rare variant and sarcomeric protein titin (*TTN*) mutation carriers are slightly higher or similar to other patients [82,83,84]. Apart from single gene mutations, multi-gene mutation carriers may have an even higher risk of HTx or death due to HF [10,36]. The majority of multi-gene mutation carriers have at least one HF related symptoms [10].

Previous genetic studies of inherited cardiomyopathies demonstrated that the different pattern of gene mutation may lead to distinct prognosis [11,36,85]. It is known that *PKP2* carriers are more likely develop ventricular arrhythmias, while a study of whole genome sequencing and transcriptome sequencing in heart transplanted ACM patients found that recessive variants in *PKP2* may lead to early-onset advanced HF [85]. Furthermore, the prognosis of homozygous mutation carriers is much worse than those carrying missense mutations. In some other conditions, however, the risk of severe HF in missense mutations and premature truncating and splice site mutations carriers demonstrates no significant difference [11].

Except for cohort studies, gene-editing animal models mimicking ACM progression also revealed that the genotype was strongly correlated with HF. Myocardium fibrosis and aseptic inflammation were observed at all stages in *DSG2* mutation mice [86]. These abnormalities may lead to cardiomyocytes’ death, ventricular dilation and HF progression.

### 4.4. Physical Activity and Exercise

There is a general consensus that exercise is an important modulator which could promote ACM’s progression and has a significant effect on the prognosis of ACM patients. Endurance exercise and competitive sports may increase the susceptibility to lethal arrhythmic events [87,88,89,90]. In fact, intense physical exercise may also aggravate and accelerate myocardial dysfunction and the progression of HF in the ACM population. In a clinical study in ACM athletes, the occurrence of biventricular dysfunction was more common in athletes than in non-athletes and mutation-positive family members in ACM. All HTx occurrences were in the athlete group, while none underwent HTx in the nonathletic group [25]. Competitive exercise is associated with biventricular cardiac dysfunction and elevation of hs-cTnT (high-sensitivity cardiac troponin T) and NT-proBNP [91]. Healthy athletes had physiological changes in myocardial compensatory hypertrophy, mild dilatation of LV and (or) RV and lower normal range of LVEF and (or) RVEF, but these parameters were significantly worse in ACM athletes [25]. The progression of fibro-fatty replacement in RV can also be accelerated because of heavy exercise [92,93]. They all suggested that the amount and intensity of exercise activity had a strong association with the degree of LV and RV function impairment. Thus, ACM patients are recommended to avoid strenuous exercise. However, lifestyle without any activity may also impair the metabolic capacity, muscle strength and mental health of patients. In addition, recreational sport participation with a small amount of activity seems to be harmless with respect to time free from VT and HF in ACM patients [94].

## 5. Prevention and Management

HF is a main cause of HTx and cardiac death in ACM. Timely and effective prevention and management are of great significance to delay or reverse the course of HF progression and improve the survival condition of the ACM patients. Management strategy should be in parallel with the ACM clinicopathological stages based on the 2019 HRS expert consensus statement of ACM [95] and 2021 ESC HF guidelines [96] (Figure 8). Upon the diagnosis of ACM, genetic testing and family screening are necessary. In addition, lifestyle modifications such as exercise restriction and a low-sodium diet are also needed. When electrical instability symptoms have been presented, consistent hemodynamic follow-up, anti-arrhythmic drug treatment and proper ICD implantation should be considered. These approaches could not only release the burden of arrhythmias, but also delay HF progression. With the aggravation of HF during the ACM’s progression, therapy regimens could add anti-HF and/or antithrombotic drugs. Surgical treatments such as implantation of ventricular assist devices, cardiac resynchronization therapy (CRT) and HTx are the final solutions to end-stage HF in ACM.

### 5.1. Prevention

There is no specific drug aimed at reversing the clinical course of HF in ACM [9], so screening and prevention at an early stage is particularly important. As mentioned above, genotype has a strong correlation with the phenotype and prognosis of ACM. Thus, genetic testing and family screening after definite diagnosis is necessary. Hemodynamic disorders can also influence the progression of HF, and therefore, careful hemodynamic follow-up shall be advised [36].

According to the limited case reports from different centers, pregnancy in ACM is safe and can be well-tolerated. Besides, repeated pregnancies do not seem to correlate with worse outcomes [36]. However, larger cohort studies in pregnant patients are still lacking. Considering that pregnancy will increase the volume overload and circulatory demand, additional risk may be added to pregnant women with ACM, especially to those patients with RV dysfunction and LV involvement. Prenatal consultation can be advised to lower the risk of HF in ACM patients [97].

### 5.2. Drug Therapy

At the early stage of HF, application of loop diuretics and aldosterone antagonists in patients with volume overload can reduce preload effectively [10]. The clinical benefit of angiotensin-converting enzyme inhibitors (ACEI) and angiotensin receptor blockers (ARB) in HF therapy has been widely accepted. In asymptomatic patients with LVEF < 45%, a combined application of ACEI/ARB can be recommended. In symptomatic patients, beta-blockers, spironolactone, ivabradine and loop diuretics may be considered. In ACM patients with LV involvement and HF, standardized therapy can be recommended based on the 2021 ESC guidelines [96]. Some patients with biventricular HF can improve, with complete or partial recovery of LV functions [16]. RV dilation and RV failure may be difficult to control with therapy. To improve RV failure in ACM, phosphodiesterase type 5 inhibitors may be used, which may improve heart contractility in acute RV failure patients [98]. Moreover, 17-beta-estradiol may also protect cardiomyocytes from cell apoptosis and fibrofatty replacement to some degree at an early stage [99].

Thromboembolic complications may also occur in ACM due to ventricular aneurysms or ventricular dilatation, etc., and long-term oral anticoagulation may be needed in some patients who develop RV or LV thrombi [100,101].

### 5.3. Defibrillator Implantation

The SCD is one of the most dangerous events, and it could occur in nearly 20% of ACM patients. An appropriate ICD implantation could lower the risk of SCD and other life-threatening arrythmias significantly. Additionally, it should be noted that the stable hemodynamic may be fundamental to reducing the risk of thrombosis and delaying HF progression. Although there is still no solid clinical evidence that ICD implantation can be beneficial for decreasing the risk of HF related death or HTx, it’s still one of the most important therapies in ACM patients. The clinical decision to pursue ICD implantation should be made by both physicians and patients. The ICD implantation criteria could be based on the risk prediction model for ventricular arrhythmias [5] and the 2019 HRS expert consensus statement [95].

### 5.4. Surgical Therapy

The effect of surgical therapy in HF treatment of ACM patients is still controversial. There is no solid evidence that surgical operations including ventriculoplasty [102], RV disarticulation [103], beating heart cryoablation [104] and left cardiac sympathetic denervation [105] can prolong the survival time of ACM patients who were at end-stage HF. LV assist devices (LVAD) and biventricular assist devices (BVAD) may also be indicated to bridge patients to HTx [9]. LVAD and BVAD support can supply steady circulation and successfully prolong the survival of ACM patients who await HTx. However, no apparent RV functional recovery or symptom improvement were reported during the bridge period [106,107,108,109]. Besides, after the implantation of LVAD, the contractile properties of the RV will decrease, and the haemodynamics disorder in RV could aggravate right side HF [110]. Thus, LVAD may not be recommended to patients with isolated RV failure and preserved LVEF. In rare cases, the CRT may be applied [97,111]. More evidence is needed to support the positive impact of CRT for HF management in patients with ACM.

The only standard therapy for advanced HF in ACM is HTx, and the 2016 International Society for Heart and Lung Transplantation heart transplant listing recommendations can be applicable for judging the indication of operation [112]. The survival condition is favorable after HTx, as the five-year survival rate was reported to be 80–90% [30,113,114]. Since the course and characteristics of HF in children and adults are basically similar, the prevention and treatment principles of adults are also applicable for children [26].

## Figures and Tables

**Figure 1 jcm-10-04782-f001:**
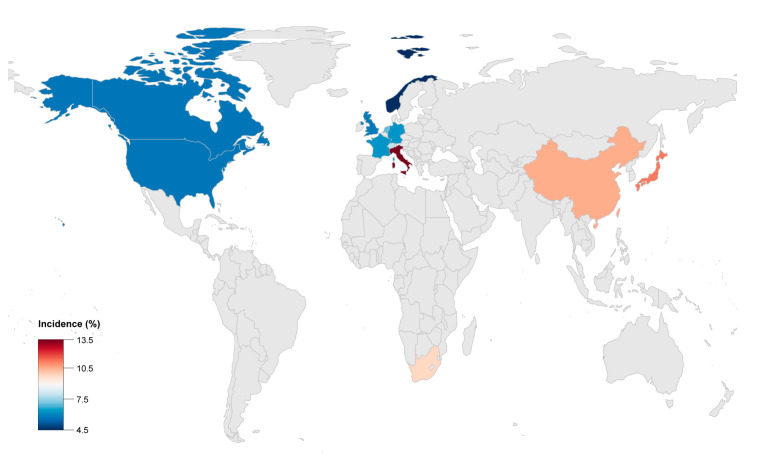
The incidence of heart transplantation/death reported from the ACM registry in different countries worldwide.

**Figure 2 jcm-10-04782-f002:**
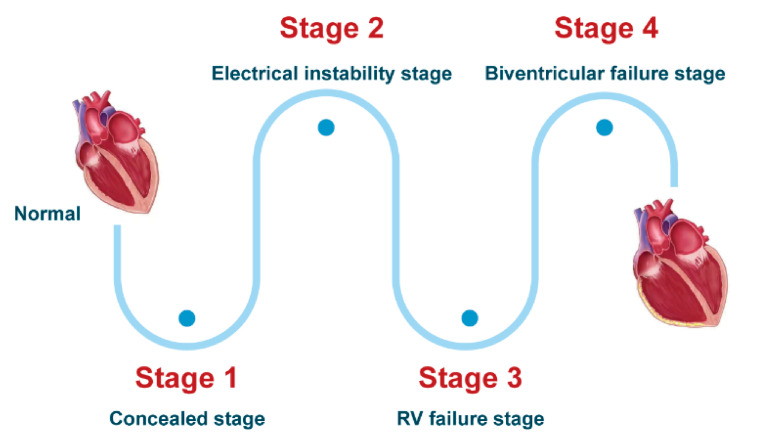
Heart failure progression in classical ACM.

**Figure 3 jcm-10-04782-f003:**
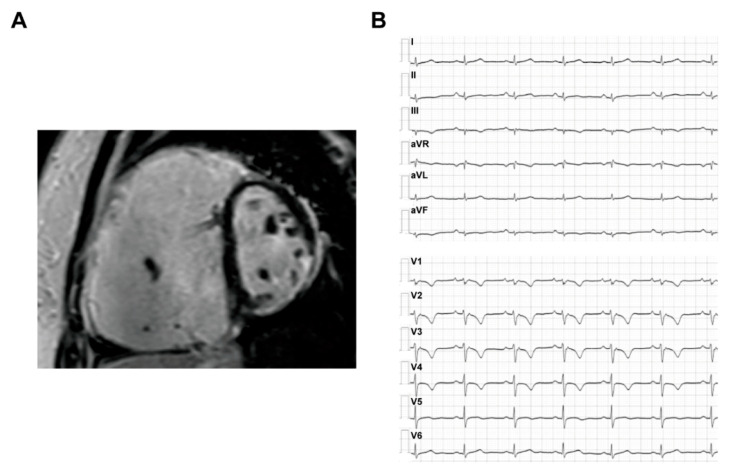
The CMR and ECG characteristics of ACM HF patients. (**A**) The CMR image of ACM HF patient; (**B**) the ECG of ACM HF patient. (Adapted with permission from Liang et al. [42]). CMR, cardiac magnetic resonance; ECG, electrocardiogram; ACM, arrhythmogenic cardiomyopathy; HF, heart failure.

**Figure 4 jcm-10-04782-f004:**
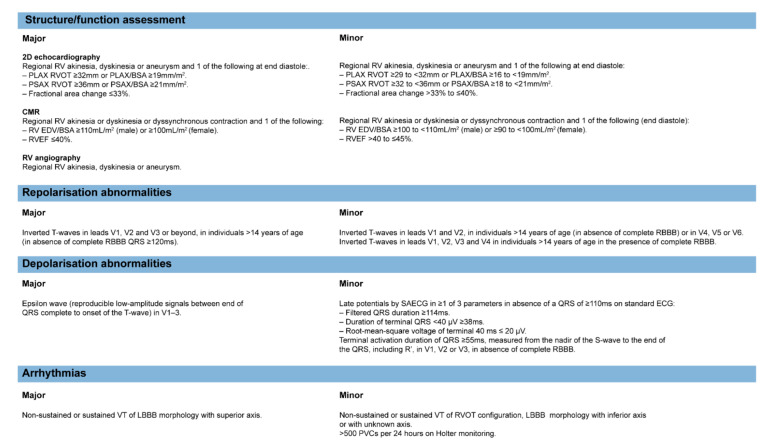
The diagnostic criteria of imaging and ECG in the 2010 Revised Task Force Criteria. 2D, two dimensional; RV, right ventricular; PLAX, parasternal long-axis; RVOT, RV outflow tract; PSAX, parasternal short-axis; BSA, body surface area; CMR, cardiac MRI; EDV, end-diastolic volume; RVEF, right ventricular ejection fraction; RBBB, right bundle branch block; SAECG, signal-averaged ECG; VT, ventricular tachycardia; LBBB, left bundle branch block; PVC, premature ventricular complex.

**Figure 5 jcm-10-04782-f005:**
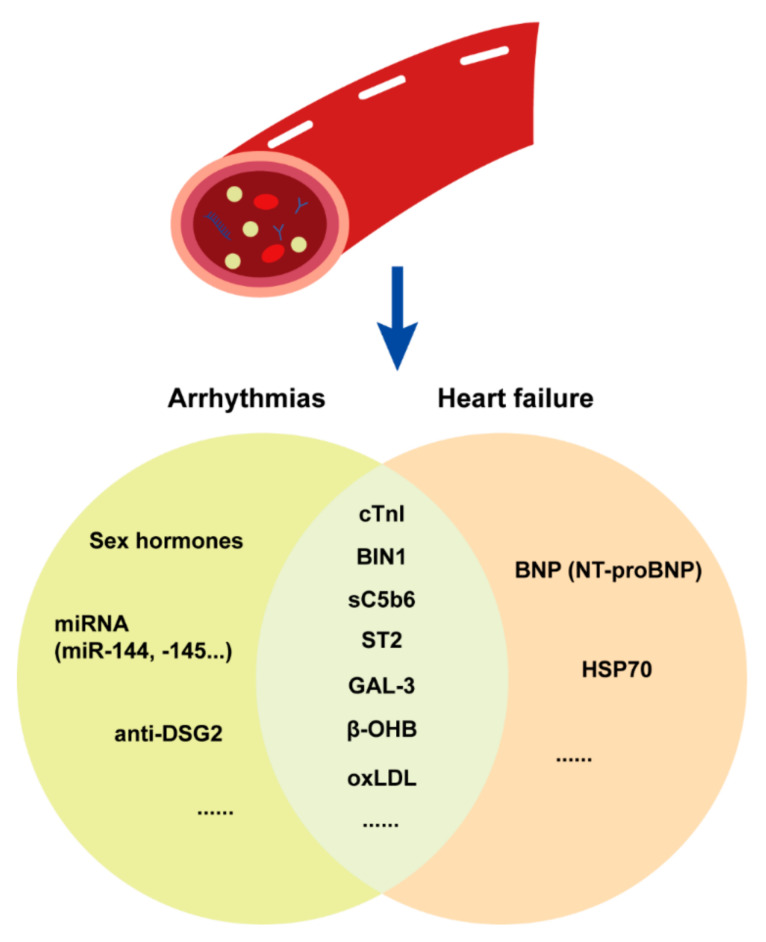
Plasma biomarker application in ACM.

**Figure 6 jcm-10-04782-f006:**
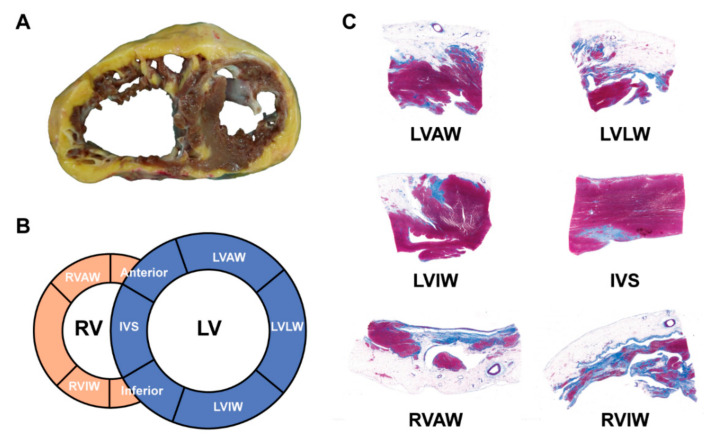
The gross morphology and histopathological characteristics of ARVC HF patients. (**A**) The explanted heart from ARVC patient; (**B**) the diagram of sampling position; (**C**) Masson staining of six representative sections. (Adapted with permission from Liang et al. [42]). LVAW, anterior wall of left ventricular (LV); LVLW, lateral wall of LV; LVIW, inferior wall of LV; IVS, interventricular septum; RVAW, anterior wall of right ventricle (RV); RVIW, inferior wall of RV.

**Figure 7 jcm-10-04782-f007:**
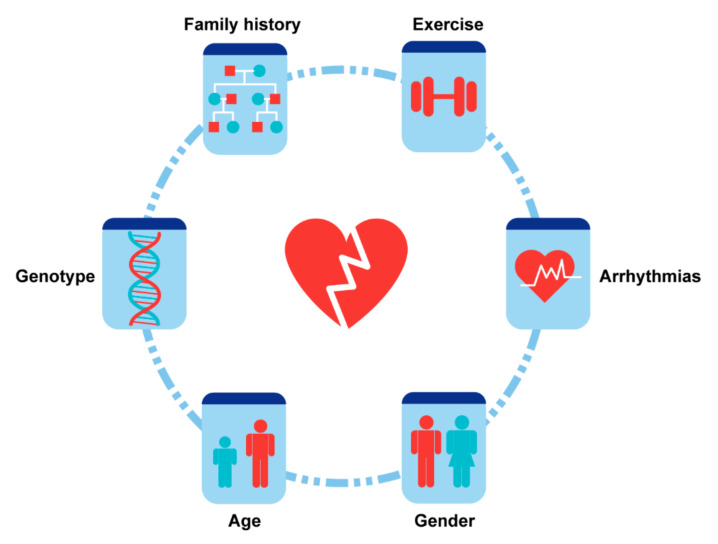
Risk factors of heart failure in ACM.

**Figure 8 jcm-10-04782-f008:**
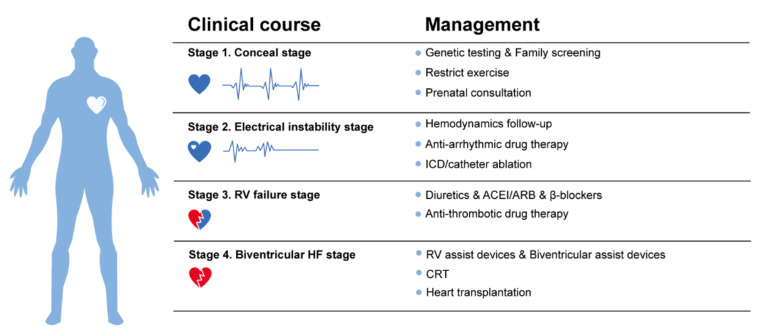
Prevention and management of heart failure in ACM. ICD, implantable cardioverter-defibrillators; ACEI, angiotensin converting enzyme inhibitors; ARB, angiotensin II receptor blockers; RV, right ventricular; CRT, cardiac resynchronization therapy.

**Table 1 jcm-10-04782-t001:** Details of heart failure studies in ACM registries.

Author	Publication Year	Study Population	Enrollment Period	Median Follow-Up (Years)	Male (%)	Endpoint	Incidence (%)	Malignant Arrythmias Events
Stefan Peters [17]	1999	121	1986–1998	12.0	52.9	1 HTx/3 death	3.3	14 RBBB
Jean-Sébastien Hulot [18]	2004	130	1977–2000	8.1	76.9	14 death	10.8	7 SCD, 17 SCA, 8 VF, 11 supraventricular arrhythmias
K Lemola [19]	2005	61		4.6	72.1	5 HTx/2 death	11.5	8 SCD, 8 SCA, 46 VT, 44 VT+LBBB
Darshan Dalal [20]	2005	100		6.0	51.0	2 HTx/1 death	3	22 SCD, 1 SCA, 51 VT+LBBB
Stefan Peters [15]	2007	313	1986–2004	8.5	62.9	2 HTx/5 death	2.2	5 SCD, 21 SCA, 96 SVT, 38 atrial arrhythmias
David A. Watkins [21]	2009	50	2004–2009	4.6	66.0	2 HTx/3 death	10	2 SCA, 41VT+LBBB
Bruno Pinamonti [22]	2011	96	1976–2008	10.7	70.8	7 HTx/6 death	13.5	3 SCA, 4 RBBB, 19 VT+LBBB, 4 atrial arrhythmias, 19 supraventricular arrhythmias
Masatoshi Komura [12]	2010	35		4.5	74.3	4 death	11.4	1 SCD
Ardan M. Saguner [23]	2013	62		7.0	67.7	3 HTx/2 death	8.1	7 SCA, 15 VF, 6 supraventricular arrhythmias
Ardan M. Saguner [24]	2014	70		5.3	67.1	5 HTx	7.1	2 SCA, 25 sustained VT, 7 VF
Jørg Saberniak [25]	2014	110			58.2	5 HTx	4.6	66 VA
Anneline S.J.M. te Riele [26]	2015	75		9.0	54.7	2 HTx/2 death	5.3	11 SCD, 8 SCA, 16 sustained VT
Judith A. Groeneweg [14]	2015	439		7.0	64.2	18 HTx/6 death	5.8	21 SCA, 96 sustained VT, 38 atrial arrhythmias
Aditya Bhonsale [11]	2015	541		6.0	58.8	8 HTx/12 death	3.7	36 SCD, 16 SCA
Cristina Gallo [27]	2016	68	1970–2014	17.0	69.1	3 HTx/4 death	10.3	3 SCD, 1 SCA
Yoshitaka Kimura [28]	2016	110		10.0	75.5	2 HTx/8 death	9.1	74 VT/VF
Aditya Bhonsale [29]	2016	502			52.6	19 HTx/53 death	14.3	34 SCD, 24 SCA, 167 sustained VT
Nisha A. Gilotra [10]	2017	289	1998–2014		50.9	15 HTx/7 death	7.6	2 SCD, 6 SCA
Thomas Gilljam [30]	2018	183	1988–2015		67.2	28 HTx	15.3	45 VT, 8 SCA, 18 atrial arrhythmias
Gabriela M. Orgeron [31]	2017	312		7.0	52.2	2 HTx/12 death	4.5	158 sustained VT, 19 VF
Saagar Mahida [32]	2019	110	2000–2015	6.4	82.7	10 HTx/3 death	11.8	3 SCD
Annina S. Vischer [9]	2019	135		7.0	61.5	5 HTx/3 death	5.9	-
Erpeng Liang [33]	2019	522	1995–2017	4.3	71.5	53 HTx/62 death	22.0	14 SCD, 136 sustained VT
Shibu Mathew [34]	2019	47	1998–2016	4.2	100	1 HTx/2 death	6.4	4 SCA, 18 sustained VT
Mikael Laredo [35]	2019	23	2003–2015	3.9	100	4 HTx	17.4	19 VT
Alexis Hermida [36]	2019	118	2006–2013	5.6	72.9	9 HTx/1 death	8.5	54 SVT/VF/SCA/SCD
Elizabeth S. DeWitt [13]	2019	32			56.3	10 HTx	31.3	5 SCA, 14 VT
L. P. Bosman [37]	2019	850		9.5	52.1	7 HTx/53 death	7.1	-

RBBB, right bundle branch block; SCD, sudden cardiac death; SCA, sudden cardiac arrest; VF, ventricular fibrillation; VT, ventricular tachycardia; VA, ventricular arrhythmia; LBBB, left bundle branch block.

**Table 2 jcm-10-04782-t002:** Fuwai classification of ACM.

	Cluster 1 (Classical ACM)	Cluster 2	Cluster 3	Cluster 4
Clinical features	Early-onset disease, ventricular arrhythmias common, usually progressive RV (and later LV) disease, large RVEDD on echo, precordial fractionation and low voltage, MACE common during follow-up	Ventricular arrhythmias common, usually progressive disease, moderate-severe LV dysfunction, precordial fractionation and low voltage	Ventricular arrhythmias common, usually progressive disease, severe LV dysfunction, large LVEDD on echo, progression to end-stage heart failure common	Ventricular arrhythmias common, usually progressive disease, severe LV dysfunction, large LVEDD and LA diameter on echo, progression to end-stage heart failure common
Histopathology	Fibrofatty infiltration RV subepicardial (early), transmural (late), LV posterior wall	Fibrofatty infiltration of RV anterior wall, LV interstitial fibrosis in full thickness with only little fat	Biventricular involvement with prominent fibrofatty infiltration, LV involvement mostly of inferior wall	LV dominant involvement, mostly of inferior wall, with prominent fibrofatty infiltration
Genetic variants	Mostly desmosomal *(PKP2, DSG2, DSC2)*	Mostly non-desmosomal *(LMNA, PLN, TMEM43, DES, CTNNA3)*	Mostly desmosomal *(DSP)* or non-desmosomal *(PLN, CTNNA3)*	No genetic variants

Adapted with permission from Firat et al. [62].

## Data Availability

Not applicable.

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
