# Peer review of "Heart Failure in Patients with Arrhythmogenic Cardiomyopathy"

_jcm, 2021, doi:10.3390/jcm10204782_

Round 1
Reviewer 1 Report
The lack of description of the methodology of the cited publications selection is a serious drawback of this manuscript. The adequate description should be provided before publication.
Author Response
The lack of description of the methodology of the cited publications selection is a serious drawback of this manuscript. The adequate description should be provided before publication.
RE: Thank you for your reminding. We added the description of the methodology of publication selection in our revised version. (Page 4, Line 53)

Reviewer 2 Report
The current manuscript consists of a review article on heart failure (HF) in patients with arrhythmogenic cardiomyopathy (ACM).
This review summarizes all the main aspects of HF in ACM patients, from epidemiology and clinical aspects to diagnosis and treatment. The Authors made great efforts for this work and the objective of this paper has been reached providing a large and accurate appraisal of all the important issues. However, shortening the text to focus on more practical messages should be also considered. In addition, several flaws can be addressed to improve this review:
- Major:
- Introduction: you stated that prevalence and severity of HF in patients with ACM are debated. However, since the aim of your review article is broader, please provide some background information (just a few lines) also on clinical course, risk stratification, prevention and treatment strategies.
- Prevalence reporting: please, be clearer on the different definitions and reporting methods across different centers.
- High prevalence differences: please, discuss why there is a so high difference between reporting in primary and tertiary care centers.
- Figure 1: please, consider to be more accurate on the contents of Figure 1. An interesting option could be to show the prevalence of HF in ACM by using a color grading.
- Prevalence of HF in ACM: please, consider shortening this paragraph. Provide some global evidence and overall interpretation of data rather than reporting each event in every single national cohort.
- Clinical presentations: please, avoid to merge different clinical presentations (e.g., electrical instability and progressive HF at lines 87-89).
- Late-onset HF: you stated “However, later-onset HF was more common in ACM, and the prognosis was worse in these patients”. Please, add the comparators to these sentences.
- HF and ventricular arrhythmias: the relationship between these two entities is a crucial point of the review. Please, discuss it more in details and be clearer at lines 101-102.
- Figure 2: please, improve this figure. The heart icons are identical for normal and HF hearts. In addition, heart failure is put at the end of the process as if it was a successive stage after biventricular failure.
- Disease progression: at page 8, you stated “Fortunately, only a small proportion of ACM patients reach this phase”. Please, note that the smaller proportion of patients at this stage can be influenced by survival bias.
- Imaging: in the paragraph on imaging, please describe the respective role of echocardiography and cardiac magnetic resonance, focusing on both diagnosis and prognostic assessment
- Diagnostic criteria: please, consider to summarize diagnostic criteria in a figure/table and even in the text before commenting on imaging or ECG abnormalities.
- Cardiac biomarkers: please, comment on the potential role of troponin and cardiac biomarkers other than those already reported. In addition, this section seems to be a list of different cardiac biomarkers and their correlation with HF in ACM. Please, provide more informed insights and expert commentary.
- Display items: beyond your display items, it could be of great interest to the readers to provide clinical cases and images about echocardiography, cardiac magnetic resonance, ECG or even histology.
- Anatomical classification: please, provide some background information on RVEF. Are available data on other indexes of right ventricular dysfunction (e.g., RVFAC)?
- Role of gender: in this paragraph, you reported worse outcomes for male patients, followed by a number of reports of similar outcomes and finally a few sentences on female sex being a risk factor. Please, summarize all this evidence and provide an overall appraisal of these findings.
- Physical activity and exercise: please, consider changing the initial sentence on physical activity since it seems to state that physical activity could be a cause of ACM (lines 318-319).
- Adverse events: please, consider adding a table with the rates of main adverse events (other than death or heart transplantation in table 1) across registries and case series to inform the readers on the magnitude of this issue.
- Practical recommendations: please, provide practical recommendations for clinicians about the management of this condition (e.g., genetic counselling, lifestyle modifications, and so on).
- HF guidelines: please, avoid referring to 2016 guidelines (370). This review should be updated incorporating latest evidence (2021 guidelines).
- Myocardial biopsy: is there a role for myocardial biopsy?
- Defibrillator implantation: please, provide more evidence on the role of defibrillator and the criteria for their implantation.
- Risk stratification: please, provide a more concise summary on risk stratification for these patients.
- Reference number: please, consider to reduce the number of references.
- Minor:
- Introduction, page 3, line 43: please, change “has” with “have”.
- Introduction, page 4, lines 45-48: please, clarify what kind of studies you are reporting (in particular, focus on the included populations).
- Prevalence of HF in ACM, page 5, line 65: please, introduce the abbreviation “ARVC/D” since it is its first occurrence in the text.
- Table 1: please, consider reporting male sex as percentage of the study sample size.
- Prevalence of HF in ACM, page 5, line 74: please, avoid the spacing in “per cent”.
- Prevalence of HF in ACM, page 5, lines 74-75: please, change “rehospitalized” into “were rehospitalized”.
- Prevalence of HF in ACM, page 5, line 75: please, substitute “dies” with “died”.
- Prevalence of HF in ACM, page 5, line 80: please, remove the word “from”.
- Clinical course and symptoms, page 8, line 111: please, introduce the abbreviation “ECG”.
- Lines 128-129: please, revise for clarity.
- Line 328: please, introduce the abbreviation “hs-cTnT”.
- Figure 5, legend: please, introduce the abbreviations used in the figure.
Author Response
The current manuscript consists of a review article on heart failure (HF) in patients with arrhythmogenic cardiomyopathy (ACM).
This review summarizes all the main aspects of HF in ACM patients, from epidemiology and clinical aspects to diagnosis and treatment. The Authors made great efforts for this work and the objective of this paper has been reached providing a large and accurate appraisal of all the important issues. However, shortening the text to focus on more practical messages should be also considered. In addition, several flaws can be addressed to improve this review:
Major:
- Introduction: you stated that prevalence and severity of HF in patients with ACM are debated. However, since the aim of your review article is broader, please provide some background information (just a few lines) also on clinical course, risk stratification, prevention and treatment strategies.
RE: Thank you for your suggestions. We added the background information of these aspects in our revised manuscript. (Page 3, Line 46)
- Prevalence reporting: please, be clearer on the different definitions and reporting methods across different centers.
RE: Thank you for your comments. According to your comments, we delete the description of the HF prevalence in each single center, and we added the clear definition of HF in two representative study. (Page 4, Line 66)
- High prevalence differences: please, discuss why there is a so high difference between reporting in primary and tertiary care centers.
RE: Thank you for your comments. We added the discussion of the causes for this high difference in our revised version. (Page 5, Line 72)
- Figure 1: please, consider to be more accurate on the contents of Figure 1. An interesting option could be to show the prevalence of HF in ACM by using a color grading.
RE: Thank you for your suggestion. We tried to find a better way to present the different prevalence of HF among different countries in Figure 1 as you advised. However, considering the different definition of HF and the different enrollment criteria in each study, the prevalence of HF and the HTx/Death risk varied a lot. Thus, we retained the original version of this figure for avoiding to confuse our readers. The readers are able to find the exact prevalence of HF in each ACM study in Table 1.
- Prevalence of HF in ACM: please, consider shortening this paragraph. Provide some global evidence and overall interpretation of data rather than reporting each event in every single national cohort.
RE: Thank you for your comments. We delete this description in our revised manuscript. (Page 4, Line 66)
- Clinical presentations: please, avoid to merge different clinical presentations (e.g., electrical instability and progressive HF at lines 87-89).
RE: Thank you for your comments. We rephrase our words in the revised version. (Page 8, Line 91)
- Late-onset HF: you stated “However, later-onset HF was more common in ACM, and the prognosis was worse in these patients”. Please, add the comparators to these sentences.
RE: Thank you for your comments. We add the comparators in our revised version. (Page 8, Line 94)
- HF and ventricular arrhythmias: the relationship between these two entities is a crucial point of the review. Please, discuss it more in details and be clearer at lines 101-102.
RE: Thank you for your comments. As you pointed out, one of the most important part of this review is to illustrate the relationship between HF and arrhythmias. According to the point of view from many valuable studies, the arrhythmias could be one of the risk factors for accelerating the HF progression in ACM. We discuss their potential relationship in details at the “Risk factor and risk stratification” part. Thus, we generally describe their relations here. We hope for your understanding.
- Figure 2: please, improve this figure. The heart icons are identical for normal and HF hearts. In addition, heart failure is put at the end of the process as if it was a successive stage after biventricular failure.
RE: Thank you for your suggestions. We improve this figure in the revised version. (Page 9, Line 112)
- Disease progression: at page 8, you stated “Fortunately, only a small proportion of ACM patients reach this phase”. Please, note that the smaller proportion of patients at this stage can be influenced by survival bias.
RE: Thank you for your reminding. We changed our statement in the revised manuscript. (Page 10, Line 126)
- Imaging: in the paragraph on imaging, please describe the respective role of echocardiography and cardiac magnetic resonance, focusing on both diagnosis and prognostic assessment
RE: Thank you for your comments. We discuss the diagnosis and prognostic role of echocardiography and CMR in the revised version. (Page 12, Line 155)
- Diagnostic criteria: please, consider to summarize diagnostic criteria in a figure/table and even in the text before commenting on imaging or ECG abnormalities.
RE: Thank you for your suggestions. We summarize the diagnostic criteria as Figure 4 in the revised version. (Page 11, Line 149)
- Cardiac biomarkers: please, comment on the potential role of troponin and cardiac biomarkers other than those already reported. In addition, this section seems to be a list of different cardiac biomarkers and their correlation with HF in ACM. Please, provide more informed insights and expert commentary.
RE: Thank you for your comments. We add our commentary of this part in the revised manuscript. (Page 13, Line 190; Page 14, Line 210)
- Display items: beyond your display items, it could be of great interest to the readers to provide clinical cases and images about echocardiography, cardiac magnetic resonance, ECG or even histology.
RE: Thank you for your advice. We provide the CMR, ECG and histology images as Figure 3 (Page 10, Line 139) and Figure 6 (Page 16, Line 223) in the revised version.
- Anatomical classification: please, provide some background information on RVEF. Are available data on other indexes of right ventricular dysfunction (e.g., RVFAC)?
RE: Thank you for comments. We emphasize the RVEF should be measured by CMR, and we added the RVFAC assessment by echo in our revised manuscript. (Page 17, Line 257)
- Role of gender: in this paragraph, you reported worse outcomes for male patients, followed by a number of reports of similar outcomes and finally a few sentences on female sex being a risk factor. Please, summarize all this evidence and provide an overall appraisal of these findings.
RE: Thank you for your comments. We summarize the evidence of these findings and discuss at the end of this part. (Page 20, Line 303)
- Physical activity and exercise: please, consider changing the initial sentence on physical activity since it seems to state that physical activity could be a cause of ACM (lines 318-319).
RE: Thank you for your comments. We changed our statement in the revised version. (Page 24, Line 374)
- Adverse events: please, consider adding a table with the rates of main adverse events (other than death or heart transplantation in table 1) across registries and case series to inform the readers on the magnitude of this issue.
RE: Thank you for your suggestions. We summarize the main adverse arrythmias related events of these studies in Table 1. (Page 6, Line 84)
- Practical recommendations: please, provide practical recommendations for clinicians about the management of this condition (e.g., genetic counselling, lifestyle modifications, and so on).
RE: Thank you for your comments. We provide the practical recommendations in our revised manuscript. (Page 25, Line 398)
- HF guidelines: please, avoid referring to 2016 guidelines (370). This review should be updated incorporating latest evidence (2021 guidelines).
RE: Thank you for your comments. We updated our statement and recommendations base on the latest 2021 guidelines in the revised version. (Page 27, Line 433)
- Myocardial biopsy: is there a role for myocardial biopsy?
RE: Thank you for your comments. We add the role of endomyocardial biopsy (EMB) in the revised version. (Page 17, Line 237)
- Defibrillator implantation: please, provide more evidence on the role of defibrillator and the criteria for their implantation. 2019 heart rythem
RE: Thank you for your comments. We add the evidence on the role of ICD and the criteria for ICD implantation in the revised version. (Page 28, Line 443)
- Risk stratification: please, provide a more concise summary on risk stratification for these patients.
RE: Thank you for your comments. We provide the summary on risk stratification in the revised version. (Page 19, Line 284)
- Reference number: please, consider to reduce the number of references.
RE: Thank you for your comments. We reduce the number of references from 157 to 119.
Minor:
- Introduction, page 3, line 43: please, change “has” with “have”.
RE: Thank you for your comments. We changed the word in the revised version. (Page 3, Line 40)
- Introduction, page 4, lines 45-48: please, clarify what kind of studies you are reporting (in particular, focus on the included populations).
RE: Thank you for your comments. The studies included in this part are all retrospective studies, and we clarify this point in the revised version. (Page 3, Line 42)
- Prevalence of HF in ACM, page 5, line 65: please, introduce the abbreviation “ARVC/D” since it is its first occurrence in the text.
RE: Thank you for your comments. We introduce the abbreviation “ARVC/D” in the revised version. (Page 4, Line 54)
- Table 1: please, consider reporting male sex as percentage of the study sample size.
RE: Thank you for your suggestions. We changed the number to the percentage in Table 1. (Page 6, Line84)
- Prevalence of HF in ACM, page 5, line 74: please, avoid the spacing in “per cent”.
RE: Thank you for your comments. We delete the part of HF prevalence in each single center. Thus, this sentence is deleted from the original version.
- Prevalence of HF in ACM, page 5, lines 74-75: please, change “rehospitalized” into “were rehospitalized”.
RE: Thank you for your comments. We delete the part of HF prevalence in each single center. Thus, this sentence is deleted from the original version.
- Prevalence of HF in ACM, page 5, line 75: please, substitute “dies” with “died”.
RE: Thank you for your comments. We delete the part of HF prevalence in each single center. Thus, this sentence is deleted from the original version.
- Prevalence of HF in ACM, page 5, line 80: please, remove the word “from”.
RE: Thank you for your comments. We delete the part of HF prevalence in each single center. Thus, this sentence is deleted from the original version.
- Clinical course and symptoms, page 8, line 111: please, introduce the abbreviation “ECG”.
RE: Thank you for your reminding. We introduce the abbreviation “ECG” in the revised version. (Page 9, Line 115)
- Lines 128-129: please, revise for clarity.
RE: Thank you for your comments. We rephrase our statement in the revised version. (Page 10, Line 132)
- Line 328: please, introduce the abbreviation “hs-cTnT”.
RE: Thank you for your reminding. We introduce the abbreviation “hs-cTnT” in the revised version. (Page 24, Line 382)
- Figure 5, legend: please, introduce the abbreviations used in the figure.
RE: Thank you for your reminding. We introduce the abbreviations used in Figure 8 in the revised version. (Page 26, Line 411)

Reviewer 3 Report
The authors of the current manuscript have tried to make an overview of the diagnostic and therapeutic approach to patients with heart failure, following arrhythmogenic cardiomyopathy, and to discuss some solved and unsolved issues related to the failing cardiac pump function in these patients.
I have the following recommendations to the authors:
Line 19-21” I suggest some modification in this sentence, like “Development of heart failure (HF) has been an under-recognized complication of arrhythmogenic ventricular cardiomyopathy (ACM) for a long time.” Or something like that.
Line 21: „Although growing attention on HF has been arisen in recent years, the systemic summary of its clinical features is still lacking.“ – this is redundant
Line 22: “The current cognition on HF in ACM progression is in the mist, and no reviews focused on this field are available until now. A comprehensive understanding of the HF phenotype in ACM will be great beneficial to clinical management and prognosis improvement.” – it has to be modified. There are clear criteria for recognition (diagnosing) of heart failure, updated in 2021, and ACM I mentioned in the current HF Guidelines there as a possible cause of HF.
Line 370: Please, update the therapy guidelines of HF according to the new ESC 2021 guidelines or the 2020 ACC/AHA guideline or other international guidelines
Line 383-386: Please, re-check the current guidelines for LVAD treatment. Candidates for this treatment should not have severe right ventricular dysfunction and/or severe tricuspid regurgitation.
Author Response
The authors of the current manuscript have tried to make an overview of the diagnostic and therapeutic approach to patients with heart failure, following arrhythmogenic cardiomyopathy, and to discuss some solved and unsolved issues related to the failing cardiac pump function in these patients.
I have the following recommendations to the authors:
Line 19-21” I suggest some modification in this sentence, like “Development of heart failure (HF) has been an under-recognized complication of arrhythmogenic ventricular cardiomyopathy (ACM) for a long time.” Or something like that.
RE: Thank you for your comments. We changed our statement in the revised manuscript. (Page 2, Line 19)
Line 21: „Although growing attention on HF has been arisen in recent years, the systemic summary of its clinical features is still lacking.“ – this is redundant
RE: Thank you for your comments. We delete this sentence in the revised version.
Line 22: “The current cognition on HF in ACM progression is in the mist, and no reviews focused on this field are available until now. A comprehensive understanding of the HF phenotype in ACM will be great beneficial to clinical management and prognosis improvement.” – it has to be modified. There are clear criteria for recognition (diagnosing) of heart failure, updated in 2021, and ACM I mentioned in the current HF Guidelines there as a possible cause of HF.
RE: Thank you for your comments. We rephrase our statement in the abstract and manuscript based on the latest 2021 guidelines. (Page 2, Line 21)
Line 370: Please, update the therapy guidelines of HF according to the new ESC 2021 guidelines or the 2020 ACC/AHA guideline or other international guidelines.
RE: Thank you for your comments. We update our statement based on the latest 2021 guidelines (Page 27, Line 433)
Line 383-386: Please, re-check the current guidelines for LVAD treatment. Candidates for this treatment should not have severe right ventricular dysfunction and/or severe tricuspid regurgitation.
RE: Thank you for your comments. We rephrase our statement of the LVAD treatment in the revised version. (Page 29, Line 461)

Round 2
Reviewer 1 Report
The manuscript entitled Heart Failure in Patients with Arrhythmogenic Cardiomyopathy is a comprehensive description of this topic and should be published without any further changes.
Author Response
Thank you for your approval.
Reviewer 2 Report
Dear Authors,
Thank you for your work on this manuscript based on reviewers’ comments. I appreciated the acceptance of many suggestions that I hope could be of help to improve your paper.
However, I still have a few major comments about this manuscript:
- Introduction: although you added a few lines on background information, main points (i.e., clinical course, risk stratification, prevention and treatment strategies) are not commented on. Please, consider being more detailed to inform readers who are not necessarily expert about this specific condition.
- Search strategy: in the introduction, you listed the source of information your review is based on. Please, note that you included medRxiv, which is a pre-print service rather than a scientific database, meaning that it may contain publications who have not undergone peer review.
- Figure 1: although your reply in the rebuttal is understandable, Figure 1 does not add no information to the text as it is. Please, consider again changing the representation of epidemiological measures by applying a color grading.
- Finally, shortening the text to focus on more practical messages should be carefully considered once again.
Author Response
Thank you for your work on this manuscript based on reviewers’ comments. I appreciated the acceptance of many suggestions that I hope could be of help to improve your paper.
However, I still have a few major comments about this manuscript:
Introduction: although you added a few lines on background information, main points (i.e., clinical course, risk stratification, prevention and treatment strategies) are not commented on. Please, consider being more detailed to inform readers who are not necessarily expert about this specific condition.
RE: Thank you for your comments. We added a more detailed introduction and background information of the clinical course, risk stratification, prevention and treatment strategies in ACM in this version. (Page 3, Line 46-54)
Search strategy: in the introduction, you listed the source of information your review is based on. Please, note that you included medRxiv, which is a pre-print service rather than a scientific database, meaning that it may contain publications who have not undergone peer review.
RE: Thank you for your comments. In order to search the articles in this field as comprehensively as we can, we used PubMed, Embase and medRxiv at first. As you mentioned, the medRxiv is a pre-print platform and the articles are not that convincing. Thus, we deleted these articles in medRxiv from further summary and analysis, and all the references included in this review were peer-reviewed and published in PubMed or Embase. We deleted “medRxiv” from our manuscript in this version to avoid misleading the readers. (Page 4, Line 59)
Figure 1: although your reply in the rebuttal is understandable, Figure 1 does not add no information to the text as it is. Please, consider again changing the representation of epidemiological measures by applying a color grading.
RE: Thank you for your comments. We improve the Figure 1 by color grading according to the incidence of cardiac death/HTx in these countries in the revised manuscript. (Page 5, Line 81) We can find the ACM HF problems are more severe in Asian and African countries compared to European or American countries through this version directly.
Finally, shortening the text to focus on more practical messages should be carefully considered once again.
RE: Thank you for your comments. We delete part of the description in “histopathological characteristics” section, which could be not useful in clinical practice. (Page 17, Line236) We delete the “Anatomical classification” section, (Page 18, Line 247) which was partly described in the “Clinical course and symptoms” and “Histopathological characteristics” section. Besides, we simplify the content of “Fuwai Classification” section. (Page 18, Line 247-255) We reduce the section of histopathological characteristics and HF classification, which are possible not that important for physicians in solving their clinical problems. We more focus on the clinical information of ACM heart failure, from prevalence, natural course/symptoms, imaging/histopathological/plasma examination, risk stratification to prevention/treatment strategy. We hope this review can provide more comprehensive and practical messages to the readers through revision.

Reviewer 3 Report
The authors of the manuscript "Heart Failure in Patients with Arrhythmogenic Cardiomyopathy" have observed the reviewer's recommendations and have significantly improved the quality of their article.
The manuscript can be published in its current version.
Author Response
Thank you for your approval.